# FROM CODE TO ACTION: HIERARCHICAL LEARNING OF DIFFUSION-VLM POLICIES

## ABSTRACT

Imitation learning for robotic manipulation often suffers from limited generalization and data scarcity, especially in complex, long-horizon tasks. In this work, we introduce a hierarchical framework that leverages code-generating vision-language models (VLMs) in combination with low-level diffusion policies to effectively imitate and generalize robotic behavior. Our key insight is to treat open-source robotic APIs not only as execution interfaces but also as sources of structured supervision: the associated subtask functions - when exposed - can serve as modular, semantically meaningful labels. We train a VLM to decompose task descriptions into executable subroutines, which are then grounded through a diffusion policy trained to imitate the corresponding robot behavior. To handle the non-Markovian nature of both code execution and certain real-world tasks, such as object swapping, our architecture incorporates a memory mechanism that maintains subtask context across time. We find that this design enables interpretable policy decomposition, improves generalization when compared to flat policies and enables separate evaluation of high-level planning and low-level control.

## 1 INTRODUCTION

The field of robotics has increasingly embraced imitation learning and the expansion of data collection as pivotal research avenues, inspired by the recent successes of generative models in language and vision domains (Kim et al., 2024; Ha et al., 2023; Team et al., 2024; Brohan et al., 2022). Unfortunately, however, the challenge of obtaining high-quality and diverse data necessary for training robots to perform a wide array of tasks remains a problem due to the need for accurate language annotations and corresponding expert demonstrations (Blank et al., 2024). On the other hand, many robotics tasks share a common trait of compositionality, which is akin to functional programming: Sophisticated programs may appear to exhibit highly complex behavior that is difficult to imitate, but they are usually compositions of simpler functions that are easy to understand. Similarly, navigating and manipulating objects can result in long-horizon, complex patterns that, when broken down into simple skills, become easy to learn. Once learned, skills can then be dynamically composed to potentially achieve greater adaptability and generalize to new tasks.

This idea is not novel; the robotics community has extensively studied pick-and-place tasks because they are fundamental building blocks for interacting with the world (Siciliano et al., 2008). Nonetheless, learning atomic skills and composing them into complex behaviors is challenging for a variety of reasons. Firstly, one needs to either rely on unsupervised learning to decompose long-horizon tasks, or assume access to labeled demonstrations for each subtask, which can be costly to obtain. Secondly, simply having access to a skill library is not sufficient when dealing with high level instructions, as they too first need to be translated into skills, which is exacerbated by the difficulty of long-horizon planning (Chen et al., 2025; Mishani et al., 2025).

To address the former, this paper builds on the insight that open-source robot control APIs can be a valuable source of data collection, as they not only provide expert demonstrations, but also come with annotations in the form of a code trace of their action. This code naturally exhibits a hierarchy of complexity and compositions of simple functions, making it well-suited for automating the collection of sub-task labels. Unlike natural language, which tends to be under-specified on the end state of an instruction Gu et al. (2023), these sub-task code labels are precise and unambiguous, making them ideal for robust concatenation. In order to utilize code as instructions for an end-to-end imitation

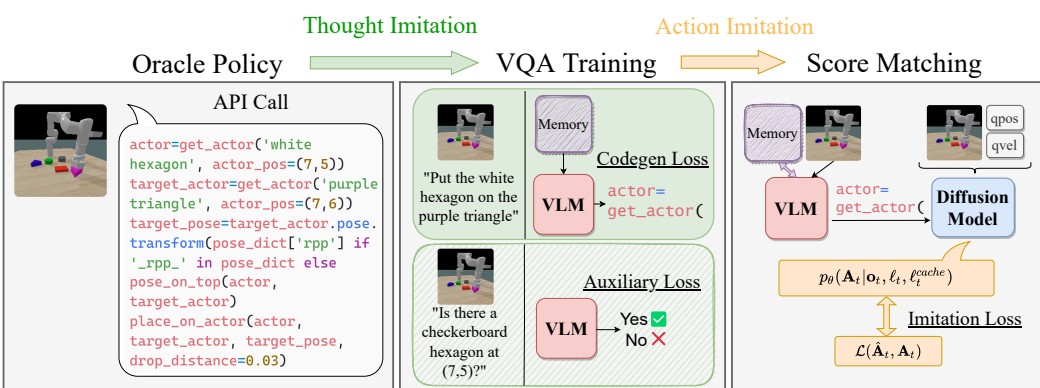

Figure 1: An illustration of our hierarchical learning approach combining thought imitation and action imitation. An oracle policy consisting of Python API calls collects demonstration data including corresponding code snippets per executed action. During the visual-question-answering (VQA) stage, a VLM is trained on the oracle demonstrations to generate the underlying API code (*codegen loss*) as well as recognize objects in the scene (*auxiliary loss*). Finally, a diffusion model, conditioned on the generated code, is trained to imitate low-level actions of the oracle.

learning system, we propose a hierarchical framework involving a code **generating** vision-language model (VLM) trained to imitate the language descriptions of API policies and a code **guided** low-level policy based on diffusion models (Chi et al., 2023) learning to dynamically map code to actions. Our training scheme is visualized in Figure 1: We first train a VLM to generate API calls from successful demonstrations of an oracle policy. Subsequently, we distill the low-level action part of the oracle policy into a custom language-conditioned diffusion policy (DP) while conditioning on VLM generated code. This ensures that any generated code trace during training mimics those observed during inference, where both models operate simultaneously. We find that this approach effectively mitigates distribution shift and improves generalization compared to a policy that relies solely on high-level task descriptions. Furthermore, by incorporating a memory mechanism into both the high- and low-level policies, we demonstrate that our model can handle non-Markovian tasks, as well as the inherently stateful nature of oracle policy code, which requires memory to function correctly.

This work serves two purposes. First, as a study of the performance of diffusion policy under various conditioning inputs: no conditioning, natural language conditioning or verifiably correct text conditioning (i.e. executable code). The second interpretation is as a method to distill existing scripted robot policies into learned policies. The applied use-case of this method is to distill a classical robotic setup which relies on many sensors and precise calibration and scripted policies into an AI-based system, which relies only a camera and robot proprioception.

**Contributions.** Our contributions can be summarized as follows:

- We introduce a novel VLM training scheme for code generation of robotic control primitives, including auxiliary losses and a memory buffer of past actions to tackle state tracking.
- We present a hierarchical framework for training code-conditioned diffusion models on VLM-labeled demonstration data, as well as a custom encoder based on learned attention pooling layers for processing multimodal conditioning information.
- We find that by accurately composing sub-tasks at inference time, our hierarchical policy generalizes better than flat variants on various tasks of the ClevrSkills benchmark.

## 2 RELATED WORK

**Language-Guided Imitation Learning.** Modern imitation learning (IL) benchmarks typically require learning a single language-conditioned policy for a variety of tasks (Mees et al., 2022; Walke et al., 2023; Haresh et al., 2024). Diffusion policies (Chi et al., 2023) offer a strong IL baseline and have since been adapted to tackle this by adding pretrained language encoders (Ha et al., 2023; Reuss et al., 2024; Li et al., 2024). Similarly, vision-language-action (VLA) models have been proposed, processing language and vision instructions through a more close integration of pretrained foundation

models into the policy. Architectural choices commonly range from using diffusion heads (Wen et al., 2024a; Liu et al., 2024b; Wen et al., 2024b) and flow matching (Black et al., 2024) to directly predicting action tokens through language (Kim et al., 2024; Zawalski et al., 2024).

**Hierarchical Policies.** Hierarchical models aim to generalize to new tasks by factorizing their action distribution into high and low-level predictions with varying choices of intermediate representations. Hierarchical diffusion models (Ma et al., 2024; Chen et al., 2024) split action generation into key-step prediction and inpainting steps, while VLM-based models have been used to predict a large variety of representations (Pan et al., 2025; Liu et al., 2024a; Stone et al., 2023; Li et al., 2025; Pan et al., 2024; Ingelhag et al., 2024) as well as natural language (Wen et al., 2025; Shi et al., 2025; Zhong et al., 2025). More closely to our work, several works have explored using code to represent policies (Xie et al., 2025; Liang et al., 2023; Li et al., 2023; Singh et al., 2023; Varley et al., 2024; Zhi et al., 2024). Typically, a pretrained (vision-)language model is leveraged to generate code corresponding to a multi-step plan, given a natural language description of a task. The focus hereby mostly lies on improving the high-level planning capabilities, whereas the low-level policy is obtained by directly executing robot API code. In our paper, code serves merely as an intermediate representation, with the goal of learning both high and low-level policies entirely through neural networks.

Akin to our paper, recent works such as HAMSTER (Li et al., 2025), HiRobot (Shi et al., 2025), Gr00t N1 (Bjorck et al., 2025) and DexVLA (Wen et al., 2025) fully realize high and low-level policies within conditional generative models. Our research diverges by focusing on the generalization performance in an idealized framework, where we obtain perfect access to subtask labels by generating code-annotated demonstration data using robot APIs. This approach precisely specifies high-level thoughts for each time step, unlike 2D path representations (Li et al., 2025), natural language (Shi et al., 2025; Wen et al., 2025) or latent thoughts (Bjorck et al., 2025). As a result, we can not only isolate the success rate of the high-level planner from the success rate of the low-level policy, but also automate data collection by directly letting the high-level planner act in the environment. The latter advantage has already been realized in the case of training non-hierarchical policies (Ha et al., 2023; Duan et al., 2024; Ahn et al., 2024).

## 3 PRELIMINARIES

**Imitation Learning** Language conditioned imitation learning aims to learn a policy $\pi_\theta : \mathcal{O} \times \mathcal{L} \to \Delta\mathcal{A}$ mapping observations $\mathbf{o}_t \in \mathcal{O}$ and task descriptions $\ell_t \in \mathcal{L}$ to a probability distribution over actions $\mathbf{A}_t \in \mathcal{A}$. More specifically, we assume to always predict a sequence of actions (*action chunk*), i.e. $\mathbf{A}_t = [\mathbf{a}_t, \mathbf{a}_{t+1}, ..., \mathbf{a}_{t+H}] \in \mathbb{R}^{H \times D_a}$, where $H$ is the prediction horizon (Zhao et al., 2023; Chi et al., 2023) and $\mathbf{o}_t = [\mathbf{X}_t^b, \mathbf{X}_t^w, \mathbf{s}_t]$ consists of image inputs $\mathbf{X}_t \in \mathbb{R}^{H' \times W \times C}$ corresponding to base and wrist cameras as well as low-dimensional proprioception features $\mathbf{s}_t \in \mathbb{R}^{D_s}$.

**Diffusion Policy.** Diffusion policy (DP) (Chi et al., 2023) parametrizes $\pi_\theta$ using diffusion models such as DDPM (Ho et al., 2020), which entails training a conditional latent variable model $p_\theta(\mathbf{A}^0|\mathbf{o}, \ell) = \int p_\theta(\mathbf{A}^{0:K}|\mathbf{o}, \ell) d_{\mathbf{A}^{1:K}}$. The latents $\mathbf{A}^{1:K}$ are noisy versions of the original data, defined by a forward noise process $q(\mathbf{A}^k|\mathbf{A}^{k-1}) = \mathcal{N}(\mathbf{A^k}; \sqrt{1-\beta_k}\mathbf{A}^k, \beta_k\mathbf{I})$ and $\beta_k > 0$. To reverse the noising process, the model is parametrized as $p_\theta(\mathbf{A}^{k-1}|\mathbf{A}^k, \mathbf{o}, \ell) = \mathcal{N}(\mathbf{A}^{k-1}; \boldsymbol{\mu}_\theta(\mathbf{A}^k, k|\mathbf{o}, \ell), \sigma_k^2\mathbf{I})$ and trained using a weighted Evidence Lower Bound (ELBO) loss (Kingma & Gao, 2023). Finally, sampling from $\pi_\theta(\mathbf{o}, \ell)$ is performed by ancestral sampling, starting from $\mathbf{A}^K \sim \mathcal{N}(\mathbf{0}, \sigma^2\mathbf{I})$ and iteratively sampling $\mathbf{A}^{k-1} \sim p_\theta(\cdot|\mathbf{A}^k, \mathbf{o}, \ell)$.

**Vision-Language Models.** Vision-language models (VLMs) are versatile models pretrained on large-scale, multimodal internet data (Liu et al., 2023). For our purposes, we assume VLMs to model a distribution $p_\phi(\ell^{out}|\mathbf{X}^b, \ell^{in})$ trained using next-token prediction. Given a single (base camera) image $\mathbf{X}^b$ and a task description $\ell^{in}$, a language suffix $\ell^{out}$ is predicted autoregressively $p_\phi(\ell^{\text{out}} \mid \mathbf{X}^b, \ell^{\text{in}}) = \prod_{t=1}^T p_\phi(l_t \mid l_1, \ldots, l_{t-1}, \mathbf{X}^b, \ell^{\text{in}})$ via a decoder-only Transformer architecture.

## 4 FROM CODE TO ACTION

To optimally facilitate thought and action imitation respectively, our training pipeline splits data generation and training into two stages, which we visualize in Figure 1. Firstly, we train a code-generating VLM on an oracle dataset generated using API calls from hard coded policies, as described

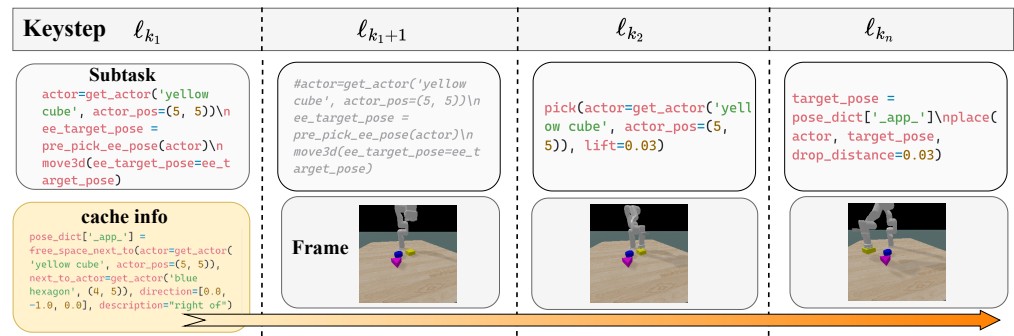

Figure 2: Illustration of a code trace on the task *PlaceNextTo*. Key-steps $\ell_{k_i}$ form unique subtask labels, while in-between steps correspond to the most recent key-step. To condition the low level policy on historical information, we extract commands that write to an internal dictionary `pose_dict` and save them to a cumulative cache (*cache info*), while the high level policy is endowed with memory by conditioning on a history $m_t$ of key-step instructions.

in Section 4.1. Using a visual-question-answering (VQA) format, the VLM is trained to predict the current action in the form of API code, given an image and a task prompt. We also introduce auxiliary losses for bounding box predictions and a memory mechanism for state tracking, which we elaborate on in Section 4.2. Secondly, we train a conditional diffusion model to predict low-level actions, given code instructions generated from the VLM, which we outline in Section 4.3.

## 4.1 DATA GENERATION

To obtain the oracle dataset $\mathcal{D}_{oracle} = \{\tau_i\}_{i=1}^N$, we utilize the ClevrSkills environment (Haresh et al., 2024), which comes with a variety of open-source scripted policies (called *solvers*) for each task. Since the policies are not perfect, we filter out any unsuccessful trajectories. Each trajectory consists of a sequence of observations, actions and language instructions $\tau = (\mathbf{o}_1, \mathbf{a}_1, \ell_1, \mathbf{o}_2, \mathbf{a}_2, \ell_2, \dots)$, where $\ell_t$ corresponds to the API code that was executed at time $t$ to produce action $\mathbf{a}_t$.

The policies (and hence the annotations) that ClevrSkills provides are hierarchical. For example, there is a `pick_move3d_place` policy, which internally uses `pick`, `move3d` and `place` policies, and utility functions such as `get_actor`. We chose to use the annotations at their most fine-grained level to provide detailed conditioning to the diffusion policy. For more details we refer to API in Appendix C.

We pre-process the API calls $\ell_t \in \tau$ into key-step instructions corresponding to the first time an API call is executed, and comment out code using the # symbol in any subsequent time-step with the same API call. We visualize one example of a code trace corresponding to a demonstration on the *PlaceNextTo* task in Figure 2.

## 4.2 CODE GENERATION VLM

**Architecture.** We build on the LLaVa framework (Liu et al., 2023), employing a Phi-3 language model backbone (Abdin et al., 2024) due to its favorable trade-off between competitive performance and efficient inference speed. Our objective is to construct a high-level VLM that maps image-valued inputs $\mathbf{X}^b$ and a natural language prompt $\ell^{in}$, which specifies the overall task, to an API call that, when executed, would lead to the completion of the current subtask. In practice, however, mapping cannot rely solely on the current observation, as most API-based policies operate in a non-Markovian regime - retaining state information such as previous object poses or task-relevant events across timesteps to ensure correct behavior in the future.

To effectively imitate such non-Markovian policies, we augment our VLM with a lightweight memory mechanism. Specifically, we implement a caching strategy that maintains a memory buffer $m_t$, which accumulates generated API calls over time. At each timestep $t$, the model appends the most

recent API call to the buffer only if it corresponds to a key-step. This memory is then incorporated into future predictions, enabling coherent, temporally-aware code generation. We'll provide a more detailed formalization of how and when this memory mechanism is used in the following paragraph.

**Training scheme.** Our VLM is trained on two different objectives: Code generation and auxiliary losses such as bounding box prediction and object recognition. For code generation, the general prompt structure combines memory information $m_t$, the task prompt $\ell^{in}$ and an optional key-step request $\ell^{key}$. The goal is to minimize the loss

$$\mathcal{L}_{code}(\phi) = -\mathbb{E}_{t \sim U([T])} \left[ p_\phi(\ell_t | \mathbf{X}_t^b, \ell^{in}, m_{t-1}) \right] - \mathbb{E}_{i \sim U([n])} \left[ p_\phi(\ell_{k_i} | \mathbf{X}_{k_i}^b, \ell^{in}, \ell^{key}, m_{k_{i-1}}) \right],$$

where $m_j := (\ell_{k_1}, \ldots, \ell_{\max\{k_i \leq j\}})$ is the memory buffer of previous key-step instructions and $\ell^{key}$ is an additional prompt (*Please give a keystep reply*). Both $m_j$ and $\ell^{key}$ are processed by the VLM by appending them to the instruction $\ell^{in}$.

**Efficient Inference** One of the main motivations behind splitting $\mathcal{L}_{code}$ into a key-step and an intermediate instruction objective is to enable two modes of inference.

- **VLM + Oracle Policy** Using the key-step mode $p_\phi(\ell_{k_i} | \mathbf{X}_{k_i}^b, \ell^{in}, \ell^{key}, m_{k_{i-1}})$ is useful for enabling tool usage (Qu et al., 2025) with the VLM. In our case, the tools are Python API calls to invoke the oracle policies on which the VLM was trained. Although a perfectly executed code trace does not result in a 100% success rate due to failure cases of the oracle policies, using the VLM in this mode gives us a robust policy, as well as a close-to-optimal metric for measuring performance of the high-level policy.

- **VLM + Diffusion Policy** The intermediate prediction $\hat{\ell}_t \sim p_\phi(\cdot | \mathbf{X}_t^b, \ell^{in}, m_{t-1})$ is used when using the code outputs merely as conditioning information for a learned low level policy. In this mode, we query the VLM at each timestep. To update $m_t$, we verify if $\hat{\ell}_t$ is a key-step request by checking for non commented-out code blocks. If this is not the case, $m_t$ is not updated. In practice, we also use this mechanism for speeding up inference: When the first $l = 20$ characters of $\hat{\ell}_t$ match a commented version of the last key-step in $m_{t-1}$, we truncate the auto-regressive generation through early stopping and use the last key-step instead. Although $l$ is a hyperparameter, we found it to only have a minimal impact on performance.

**Object detection.** To enhance the understanding of object locations within a scene, we introduce an auxiliary loss. We simplify the bounding box representation by dividing images into a $10 \times 10$ grid and assigning objects to the nearest patch. Although this approach may compromise some accuracy, our early experiments indicated that predicting two integer values, rather than multiple digits, offered greater robustness while maintaining performance. Consequently, for each image $\mathbf{X}^b$, we obtain a set of bounding boxes $\{(x_i, y_i)\}_{i=1}^k$. These bounding boxes are then utilized to generate a VQA format, where we query the VLM to determine if a randomly selected object is present at a specific location $(x_i, y_i)$. Additionally, we ask the VLM to directly predict $(x_i, y_i)$ based on a given object description. While the prediction of bounding boxes is not directly queried for during inference, it is still utilized when generating code instructions. For example, in the API function `get_actor()`, the location of the actor (object) in the image is used to disambiguate between actors with identical descriptions.

## 4.3 HIERARCHICAL DIFFUSION POLICY

For the low level part of our hierarchical model, we choose to train a custom language-conditioned diffusion policy architecture (Chi et al., 2023). The main modifications come from the need to encode lengthy code instructions, as well as the need to enable conditioning on a consistent memory buffer. The overall architecture is visualized in Figure 3. To encode the code instructions (*task info*) and memory instructions (*cache info*), we use a frozen T5 language model (Raffel et al., 2020), which processes each respectively and produces a sequence of token embeddings. While one could feed the entire history $m_t$ of memory into the policy at each timestep, we instead opt to preprocess $m_t$ into a single prompt $\ell^{cache}$ which only contains information about stored variables that are relevant for future frames (for details, see Appendix E). For example, in the visualized trajectory of Figure 2 there

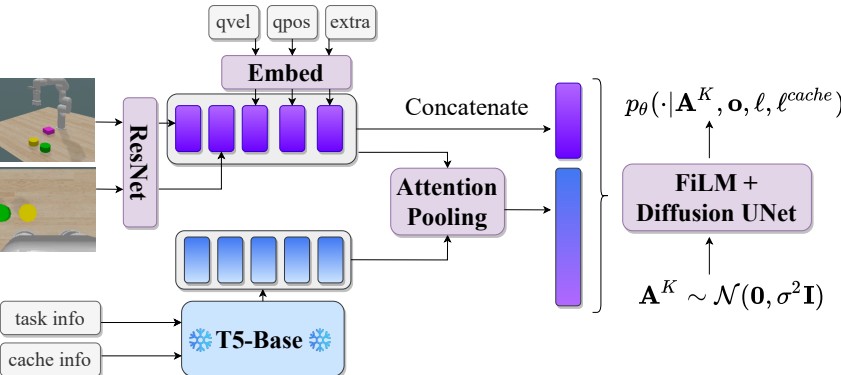

Figure 3: The low level policy is conditioned on proprioception, base and wrist camera images, as well as Python code in the form of task info for code corresponding to the current instruction and cache info for state tracking. Observation embeddings are treated as tokens and cross-attend to language embeddings using an attention pooling mechanism.

is exactly one such instruction which typically occurs at the beginning. The motivation behind this is to allow for greater generalization, since conditioning on a long history of observations can lead to overfitting to specific trajectories, reducing the model's ability to generalize to novel situations. For the same reason, we do not provide the overall task description $\ell^{in}$, but force the low level policy to rely only on subtask code instructions.

In addition to language embeddings, we use a lightweight vision encoder based on a standard ResNet-18 to process base and wrist cameras, as well as linear embedding layers for proprioception and extra information corresponding to the gripper state. We treat proprioception and image embeddings as a single token, respectively, and combine them with the language tokens using an attention pooling layer, consisting of several cross-attention blocks. The purpose of the pooling mechanism is to aggregate token-level language embeddings and arrive at a fixed-dimensional embedding, which can then be fed into a diffusion UNet head (Chi et al., 2023) with FiLM embeddings (Perez et al., 2018). Finally, to train the diffusion head, we employ DDPM with a custom loss weighting inspired by (Kingma & Gao, 2023), using $\epsilon$-prediction with a sigmoid$(-\lambda + 2)$ ELBO weighting and a cosine noise scheduler.

During training, we modify the trajectories in $\mathcal{D}_{oracle}$ to better match the distribution encountered at inference time. Specifically, we replace the oracle code instructions $\ell_t$ with generated code instructions $\hat{\ell}_t \sim p_\phi(\cdot | \mathbf{X}_t^b, \ell^{in}, m_{t-1})$. As we will show in section 5.2, training the low level policy on these generated high-level instructions leads to substantial improvements in overall performance.

Overall, the hierarchical policy is instantiated as

$$p_{\theta,\phi}(\mathbf{A}_t | \mathbf{o}_t, \ell) = p_\theta(\mathbf{A}_t | \mathbf{o}_t, \ell_t, \ell_t^{cache}) p_\phi(\ell_t, \ell_t^{cache} | \mathbf{X}_t^b, \ell, m_{t-1}),$$

where $\mathbf{A}_t$ is an action chunk of size 8. We choose to run $p_\phi$ at every step for two reasons. Firstly, the memory $m_t$ needs to be updated alongside the execution of $\mathbf{A}_t$. Secondly, we found that blind execution of an action chunk can lead to detrimental performance when the action chunk spans multiple subtasks. To mitigate this, we can stop execution of $\mathbf{A}_t$ whenever a generated instruction $\ell_k, k \in \{t, \dots, t+8\}$ contains a key-step command and regenerate a new chunk $\mathbf{A}_k$.

## 5 EXPERIMENTS

We evaluate our method on various tasks of the ClevrSkills benchmark (Haresh et al., 2024). Aside from open-source oracle solvers, which allow training our code generating VLM, ClevrSkills is built to benchmark compositional reasoning and generalization to higher level tasks. In section 5.1 we provide our main results where we aim to train a single hierarchical multitask policy and compare it to a flat baseline as well as an oracle-based baseline, whereas in section 5.2 we analyze design choices such as action chunking regeneration, dataset generation strategies and data scaling properties of our method.

Table 1: A performance comparison per task and low level training dataset. Mean success rates (%) and standard deviations are shown, computed over 64 seeds with 2 runs each.

| Task | Task Prompt Only (DP) | | | VLM+DP | | | VLM+Oracle |
|---|---|---|---|---|---|---|---|
| | L0 | L1 | L0+L1 | L0 | L1 | L0+L1 | |
| PlaceNextTo | $21.9_{\pm2.2}$ | $7.0_{\pm2.3}$ | $25.8_{\pm2.3}$ | $55.4_{\pm0.8}$ | $10.2_{\pm2.4}$ | $\mathbf{66.1}_{\pm1.1}$ | $83.1_{\pm3.7}$ |
| PlaceOnTop | $14.1_{\pm2.2}$ | $0.0_{\pm0.0}$ | $17.2_{\pm1.7}$ | $29.0_{\pm0.9}$ | $31.3_{\pm6.3}$ | $53.1_{\pm4.2}$ | $75.0_{\pm1.0}$ |
| Topple | $93.0_{\pm1.1}$ | $9.3_{\pm0.0}$ | $94.5_{\pm0.8}$ | $94.5_{\pm0.8}$ | $9.4_{\pm9.4}$ | $\mathbf{99.0}_{\pm1.0}$ | $100_{\pm0.0}$ |
| Push | $74.2_{\pm5.6}$ | $2.3_{\pm0.8}$ | $69.5_{\pm3.9}$ | $87.4_{\pm1.6}$ | $0.0_{\pm0.0}$ | $85.9_{\pm0.0}$ | $91.5_{\pm1.5}$ |
| SingleStack | $0.0_{\pm0.0}$ | $14.1_{\pm3.1}$ | $15.6_{\pm3.1}$ | $0.0_{\pm0.0}$ | $22.6_{\pm0.8}$ | $\mathbf{43.9}_{\pm4.7}$ | $81.5_{\pm1.5}$ |
| StackTopple | $0.0_{\pm0.0}$ | $0.0_{\pm0.0}$ | $0.0_{\pm0.0}$ | $0.0_{\pm0.0}$ | $17.2_{\pm1.6}$ | $\mathbf{38.9}_{\pm3.1}$ | $71.0_{\pm1.0}$ |
| PushToTarget | $2.3_{\pm1.1}$ | $30.4_{\pm5.5}$ | $8.6_{\pm2.3}$ | $0.8_{\pm0.8}$ | $87.5_{\pm1.6}$ | $82.5_{\pm7.5}$ | $75.3_{\pm0.3}$ |
| **Unseen in L0+L1** | | | | | | | |
| Pick | $35.2_{\pm5.6}$ | $0.0_{\pm0.0}$ | $35.9_{\pm1.6}$ | $59.0_{\pm3.6}$ | $67.1_{\pm0.0}$ | $\mathbf{78.0}_{\pm3.0}$ | $87.0_{\pm1.0}$ |
| ReverseStack | $0.0_{\pm0.0}$ | $0.0_{\pm0.0}$ | $0.0_{\pm0.0}$ | $0.0_{\pm0.0}$ | $21.8_{\pm4.7}$ | $\mathbf{41.4}_{\pm2.4}$ | $80.0_{\pm1.0}$ |
| NovelNoun | $14.8_{\pm1.1}$ | $0.0_{\pm0.0}$ | $14.8_{\pm1.1}$ | $26.5_{\pm1.5}$ | $26.5_{\pm7.5}$ | $\mathbf{50.7}_{\pm0.8}$ | $63.4_{\pm6.6}$ |
| Average | 25.55 | 6.31 | 28.19 | 35.24 | 29.36 | **63.95** | 80.78 |

## 5.1 MULTITASK BENCHMARK

**Setup.** To evaluate the performance of our hierarchical policy, we are not only interested in assessing general success rates per task, but also in the quantification of generalization through composing simpler subtasks to achieve new behaviors. For this purpose, we slightly deviate from the taxonomy of the compositionality of tasks introduced in ClevrSkills and simplify the benchmark into **L0** and **L1** tasks, corresponding to primitive behaviors and more complex, long horizon tasks respectively. The tasks in **L1** are chosen such that they can be achieved by composing multiple subtasks of **L0** together (for details, refer to (Haresh et al., 2024) Appendix A). To be precise, we include the tasks *PlaceNextTo*, *PlaceOnTop*, *Topple* and *Push* into the **L0** dataset, whereas the tasks *SingleStack*, *StackTopple* and *PushToTarget* are part of the **L1** datasets. Aside from *Topple* and *Push*, all tasks have 3 objects chosen at random from a collection of 32 different combinations of colors and shape.

We first generate 500 trajectories for each task of the entire ClevrSkills suite to train our high level policy. Here, we also include additional tasks such as *Pick*, *ReverseStack* and *NovelNoun* which we hold out from the training set of the low level policy as they are mostly testing language understanding and can be readily solved by reusing behaviors from **L0** and **L1** tasks mentioned above. For the low level policy, we generate 2000 trajectories for each task and we train separate policies for the **L0**, **L1** and combined **L0+L1** datasets respectively. As a comparison, we also train a flat diffusion policy with the same architecture, where language conditioning is set to $\ell_t = \ell, \ell_t^{cache} = \ell$, i.e. we replace the low level commands with identical high level *natural language* descriptions of the task.

**Results.** Table 1 shows success rates per task, separated by training dataset of the low level policy, for hierarchical (**VLM+DP**) and flat (**DP**) variants, as well as the performance of **VLM+Oracle** which is obtained by executing the key-step policy $p_\phi(\ell_{k_i}|\mathbf{X}_{k_i}^b, \ell^{in}, \ell^{key}, m_{k_{i-1}})$. The flat variant only receives the task prompt $\ell^{in}$ in the form of natural language. Overall, we find that using code instructions generated through the VLM is highly beneficial, with success rates improving across all tasks. We observe that this holds even when there is only a small overlap across tasks. For example, this can be seen when comparing success rates on the **L0** dataset with a flat variant. Here, *PlaceNextTo* sees the biggest improvement in performance with a greater than 30% increase, while the only shared primitive with other tasks is picking up the correct object, which is also found in *PlaceOnTop*. Similarly, *Push* does not share any primitives with other **L0** tasks, but still benefits from the decomposition of subtasks.

When comparing the performance of training on a the combination **L0+L1** with training on only one dataset respectively, we see that the hierarchical policy can readily reuse instructions from lower level tasks to solve longer horizon tasks. This is mainly pronounced in stacking tasks, which require chaining together *PlaceOnTop* multiple times and optionally using the *Topple* skill at the right time. However, it is interesting that zero-shot generalization of the low level to solve stacking remains challenging, with a success rate of 0% when trained only on **L0**. In this case, while the policy correctly executes the start, it tends to fail at lifting blocks high enough toward the end of

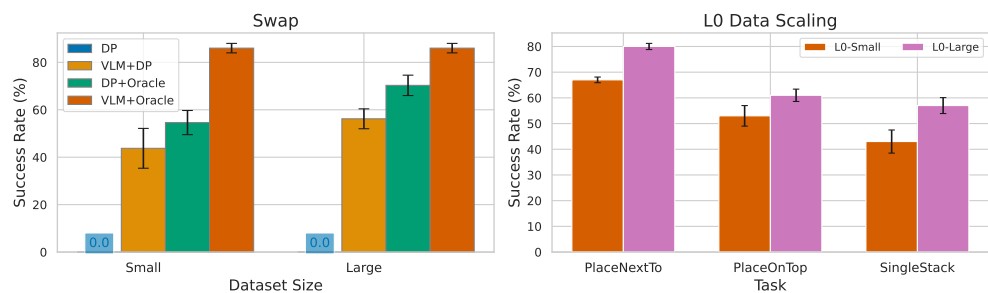

Figure 4: Left: Success rates on *Swap*, a non-Markovian task, divided into small and large datasets used to train the low level policy. Right: Success rates on pick and place tasks when training on a small and a large number of demonstrations for *PlaceNextTo* and *PlaceOnTop*.

the trajectory. Overall, to the best of our knowledge, our model is the first to generalize across these 10 tasks of the ClevrSkills suite, achieving average success rates exceeding 50%, significantly outperforming the original benchmark baseline ( Haresh et al. (2024), Table 3). Finally, we find that despite the smaller dataset, our hybrid **VLM+Oracle** policy performs strongly. This allows zero-shot generalization of the low level policy to tasks that were not seen during training, such as *Pick*, *ReverseStack* and *NovelNoun*.

**Non-Markovian Swapping**   In Table 1, all tested tasks are solvable using Markovian low-level policies. [1] To explicitly test the memory capabilities of our hierarchical policy, we train on small and large datasets of 1000 and 2000 trajectories of swapping two objects respectively. This is a challenging non-Markovian task with many subtasks, as the robot needs to (i) first remember the position of one object, (ii) pick and place it onto a free position, (iii) save the position of the other object before moving it onto the remembered initial position and (iv) pick and place the second object on the last remembered position. We note that the actual number of subtasks is closer to 12, as each moving, picking and placing instruction form their own subtasks.

We compare a flat variant trained on natural language (DP), our hierarchical policy (VLM+DP), the high level policy (VLM+Oracle) and a modified version of the task (DP+Oracle). The latter automatically calls an oracle function for computing initial positions and inserts it into the natural language task prompt. This equates to evaluating our low level policy on a Markovian version of the task. Figure 4 shows success rates for each method. As expected, DP without any high level thoughts or additional information fails regardless of training dataset size. Similar to the Markovian tasks, letting the VLM execute its generated thoughts yields the strongest performance, while learning the low level actions requires more trajectories in terms of scaling. We also observe that giving the low level policy sufficient information yields better performance than relying on the VLM. We hypothesize that this is due to the VLM failing at advancing to the next subtask if the low level policy goes out of distribution.

**Data Scaling**   We further investigate whether scaling the number of trajectories in the **L0** dataset proportionally enhances generalization to **L1** tasks. As shown on the right side of Figure 4, we evaluate a larger variant of the **L0** dataset, which includes twice the number of demonstrations for the *PlaceNextTo* and *PlaceOnTop* tasks. Our results reveal not only improved performance on

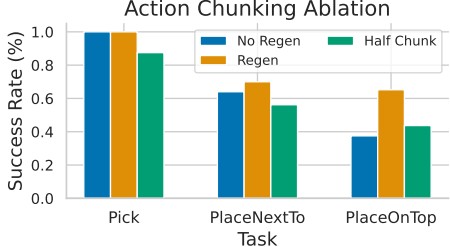

Figure 5: A comparison of different action chunking strategies during inference. *No Regen* corresponds to always executing the full predicted chunk, whereas *Regen* generates a new chunk when the VLM predicts a new key-step instruction.

---

[1]We note that this technically does not hold for the high level policy due to internal variables such as target poses, which have to be stored in memory $m_t$ even when the task itself is Markovian, see Figure 2.

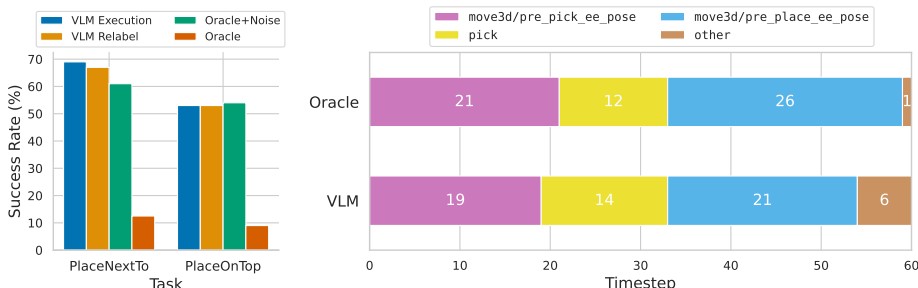

Figure 6: Comparison of different subtask labeling strategies when training on *PlaceNextTo* and *PlaceOnTop*. Left: Success rates by task and dataset. Right: Timeline comparison of instructions per timestep. The VLM tends to predict pick instructions earlier, causing performance issues when training on oracle thoughts.

these specific **L0** tasks, but also a notable increase in success rates on the more complex stacking task - despite the number of stacking demonstrations remaining constant. This cross-task improvement provides compelling evidence for compositional generalization, suggesting that the VLM effectively learns to decompose long-horizon tasks into reusable, transferable primitives.

## 5.2 ABLATIONS

**Dataset Generation** In Figure 6 we analyze the impact of different strategies for generating the thoughts $\ell_t$ for training the low level policy. We find that directly using trajectories from $\mathcal{D}_{oracle}$ leads to significantly lower performance. We hypothesize that this is due to a small mismatch in the time at which various subtasks are predicted by the VLM, compared to the start and end times of subtasks when following the oracle policy. However, we found that augmenting oracle thoughts by randomly shifting the start and end times of subtasks by up to 3 steps can mitigate this issue. Our choice of using the VLM to relabel the thoughts performs similarly on *PlaceOnTop*, while slightly better on *PlaceNextTo*. Finally, we also tested using the **VLM+Oracle** policy to generate a completely new demonstration dataset $\mathcal{D}_{exec}$, which yields the best performance on average while being more costly.

**Action Chunking** As outlined in section 4.3, we regenerate action chunks whenever a new subtask instruction is predicted by the VLM. In Figure 5, we demonstrate the performance of the hierarchical policy with and without this regeneration mechanism when trained on a dataset of 2000 *Pick*, *PlaceNextTo* and *PlaceOnTop* trajectories. We find that the regeneration becomes important when there are many subtasks to be chained together. In *Pick*, which consists of only two subtasks (moving to a pose and picking up), both methods achieve a success rate of 100%. On the other hand, both *PlaceNextTo* and *PlaceOnTop* see a decrease in performance when always executing the full action chunk. In *PlaceOnTop* this is especially pronounced, as the information on which object to place only becomes available after picking up the first object. (In *PlaceNextTo* this is not the case as the API solver precomputes free space next to objects of interests and saves it in memory). We also test halfing the prediction horizon (without regeneration), but find that it generally worsens performance.

## 6 CONCLUSION

In this work, we introduced *From Code to Action*, a hierarchical framework that integrates code-generating vision-language models with code-guided low-level policies to enable compositional generalization in robotic manipulation tasks. Our approach leverages the inherent structure of open-source API policies, allowing for automatic data collection without the need for manual subtask annotations. We investigate whether such code can serve as effective subtask supervision and demonstrate that a VLM, when provided with an appropriate memory buffer, can reliably predict the corresponding API code. Building on this, we develop a diffusion policy conditioned on the VLM-generated code and show that it significantly outperforms a flat policy baseline. Notably, our system exhibits strong signs of compositional generalization, with performance on long-horizon tasks improving as the number of training examples on simpler tasks increases.

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

## A    LIMITATIONS

Our experiments are limited to simulation only and limited to the API of the ClevrSkills benchmark. Future work includes real-world deployment as well as testing the approach on different open-source APIs. Furthermore, our low-level policy vision encoder is trained from scratch and thus naturally limited in generalization. It remains to be explored if large pretrained policies equally benefit from the hierarchical architecture proposed in this paper.

## B    LLM USAGE

We used LLMs solely for editorial assistance, including rewriting and restructuring sentences to improve clarity and readability. All scientific ideas, experimental designs and analyses are entirely our own and were conducted without the aid of LLMs. No content was generated by LLMs beyond linguistic refinement.

## C    API DESCRIPTION

Below is a description of the API which the ClevrSkills oracle uses to solve the tasks used in this paper. The VLM is trained to mimic the use of this API, and it is used as conditioning for the diffusion policy.

```python
# *************  Utility function API *************

def get_actor(
    actor: str,
    actor_pos: Optional[Tuple[int, int]] = None
) -> sapien.ActorBase:
    """
    :param actor: The name of the actor. The name is matched to
    ↪    the names
    of actors in the scene using Bleu score.
    :param actor_pos: Optional position of the actor in the
    ↪    observation
    image, relative to a coarse 10 by 10 grid. This can be used to
    disambiguate when there are multiple identical actors.
    :return: The actor which matches the description most closely.
    """

def get_pose(actor: sapien.ActorBase) -> sapien.Pose:
    """
```

```python
        :param actor: A Sapien actor.
        :return: The pose of the actor .
        """

    def free_space(actor: sapien.ActorBase) -> sapien.Pose:
        """
        :param actor: The actor to be put in free space.
        :return: A pose for actor in free space.
        """

    def free_space_next_to(
        actor: sapien.ActorBase,
        next_to_actor: sapien.ActorBase,
        direction: List,
        description: str
    ) -> sapien.Pose:
        """
        :param actor: the actor to be placed.
        :param next_to_actor: the actor to be placed next to.
        :param direction: direction (list of floats) where to
        place actor relative to next_to_actor.
        :param description: Natural language description of the
        ↪    direction
        (does not influence returned pose).
        :return: A pose for actor, next to next_to_actor, in free
        ↪    space.
        """

    def pre_pick_ee_pose(actor: sapien.ActorBase) -> sapien.Pose:
        """
        :param actor: The actor to be picked.
        :return: End-effector pose to move to, to perform a picking
        ↪    operation.
        """

    def pre_place_ee_pose(
        actor: sapien.ActorBase,
        target_pose: sapien.Pose
    ) -> sapien.Pose:
        """
        :param actor: actor to be place. Assumed to be grasped by the
        ↪    agent.
        :param target_pose: The pose to place the actor in.
        :return: the pose where end-effector should move to place the
        actor at target_pose. It is assumed that the EE is currently
        ↪    holding
        the actor.
        """
```

```python
def pre_push_pose(
    actor: sapien.ActorBase,
    topple: bool = False,
    target_pose: sapien.Pose = None,
) -> sapien.Pose:
    """
    :param actor: The actor to be pushed
    :param topple: When true, the returned pose will be closer to
    the top of the actor, because the goal is to push-to-topple.
    :param target_pose: The target to push towards. Used to
    ↪    compute
    the pushing direction.
    :return: the pose that the end-effector should move in order
    to push actor towards the target_pose.
    """

def pose_on_top(
    actor: sapien.ActorBase,
    target_actor: sapien.ActorBase
) -> sapien.Pose:
    """
    :param actor: the actor to be placed on target_actor.
    :param target_actor: The target actor.
    :return: a pose where actor is on top of target_actor.
    """

def towards_pose(
    src_pose:sapien.Pose,
    dst_pose:sapien.Pose,
    alpha:float=0.5
) -> sapien.Pose:
    """
    :param src_pose: Pose of source actor.
    :param dst_pose: Pose of destination actor.
    :param alpha: Blending coefficient between poses.
    :return: Blended position between src_pose and dst_pose.
    The orientation of src_pose is used.
    This function is used to compute how to push source actor
    towards destination actor.
    """

# ************ Policies API ************

def move3d(
    ee_target_pose: sapien.Pose = None,
    match_ori: bool = False,
    vacuum: bool = False,
    extend_bounds: float = 0.01,
```

```python
        check_done: bool = True,
    ) -> Move3dSolver:
        """
        :param ee_target_pose: the target pose of the end-effector.
        :param match_ori: Whether the orientation of the
        ↪   ee_target_pose
        must be matched.
        :param vacuum: Whether to turn vacuum gripper on or off during
        ↪   moving.
        :param extend_bounds: By how much to extend the bounds of the
        ↪   grasped
        actor (in meters) in order to avoid collections.
        :param check_done: whether the solver should check and
        ↪   self-report
        that it has completed. In most cases you want to set this to
        ↪   True.
        :return: A solver (policy) to move the end-effector to the
        ↪   specified
        pose.
        """

    def touch(
        actor: sapien.ActorBase,
        push: bool = False,
        topple: bool = False
    ) -> TouchSolver:
        """
        :param actor: The actor to be touched, pushed or toppled.
        :param push: Whether to push.
        :param topple: Whether to topple. Toppling takes priority over
        ↪   pushing.
        :return: A solver (policy) to touch/push/topple the actor.
        """

    def pick(actor: sapien.ActorBase, lift=0.1):
        """
        :param actor: The actor to be picked.
        :param lift: How much to lift the actor above the initial pose
        ↪   at
        pickup.
        Without lifting a bit, actors could be pushed off the gripper
        during horizontal transport.
        :return: A solver (policy) to pick the actor.
        """

    def place(
        actor: sapien.Actor,
        target_pose: sapien.Pose,
```

```
153     match_ori_2d: bool = False,
154     drop_distance: float = 0.02,
155 ) -> PlaceSolver:
156     """
157     :param actor: Actor to be placed.
158     :param target_pose: Absolute pose to place the actor.
159     :param match_ori_2d: Match z-axis rotation of target_pose?
160     :param drop_distance: The actor will be dropped from this
        ↪   height
161     relative to target (in meters).
162     :return: A solver (policy) to place the actor in target_pose.
163     """
164
165 def place_on_actor(
166     actor: sapien.Actor,
167     target_actor: sapien.Actor,
168     target_pose: sapien.Pose,
169     match_ori_2d: bool = False,
170     drop_distance: float = 0.02,
171 ) -> PlaceOnActorSolver:
172     """
173     :param actor: The actor to be placed
174     :param target_actor: The actor to-be-placed-upon
175     :param target_pose: pose (relative to target_actor)
176     :param match_ori_2d:  Match z-axis rotation of target_pose?
177     :param drop_distance: From what distance to drop the actor (in
        ↪   meters).
178     :return: A solver (policy) to place the actor on target_actor
        ↪   in
179     target_pose.
180     """
181
182
183 def push_along_path(actor: sapien.ActorBase, target_pose:
    ↪   sapien.Pose) -> PushAlongPathSolver:
184     """
185     :param actor: The actor to be pushed.
186     :param target_pose: The pose to be pushed towards.
187     :return: A solver (policy) to push actor to target_pose,
188     while avoiding collisions
189     """
190
```

## D   DETAILS: VLM PERFORMANCE ON CLEVRSKILLS

In Table 2 we show the success rate of the **VLM+Oracle** policy on the full ClevrSkills task suite aside from the tasks that require multimodal input prompts. Furthermore, we provide the number of average actions, which denotes the average number of API policy calls which are invoked to solve the task. Each task was evaluated on 100 random seeds.

Table 2: VLM + scripted policies: performance on a variety of ClevrSkills tasks

| Task Name | Level | Success (%) | Avg. #Actions |
|---|---|---|---|
| Pick | 0 | 99 | 2 |
| Place on top | 0 | 84 | 2.4 |
| Place next to | 0 | 96 | 2 |
| Rotate | 0 | 83 | 3 |
| Throw at | 0 | 71 | 3 |
| Throw to topple | 0 | 91 | 3 |
| Touch | 0 | 94 | 1.9 |
| Push | 0 | 93 | 2.8 |
| Topple | 0 | 100 | 2.9 |
| Pick and place on top | 1 | 79 | 5.3 |
| Pick and place next to | 1 | 96 | 4.2 |
| Follow_order | 1 | 83 | 5.2 |
| Follow_order_and_restore | 1 | 55 | 8.4 |
| Neighbour | 1 | 50 | 7.2 |
| NovelAdjective | 1 | 31 | 4.6 |
| NovelNoun | 1 | 58 | 4.1 |
| NovelNounAdjective | 1 | 56 | 4.2 |
| Rotate and restore | 1 | 72 | 4.9 |
| Rotate symmetry | 1 | 58 | 5.9 |
| Stack | 1 | 90 | 7.9 |
| Stack in reversed order | 1 | 79 | 7.5 |
| Sort by texture | 1 | 41 | 8.2 |
| Swap | 1 | 84 | 11.2 |
| Throw onto | 1 | 100 | 2 |
| Balance scale | 2 | 44 | 10.8 |
| Stack sorted_by_texture | 2 | 57 | 9.5 |
| Stack and topple | 2 | 81 | 9.9 |
| Swap by pushing | 2 | 7 | 9.8 |
| Swap and rotate | 2 | 83 | 11.3 |
| Throw and sort | 2 | 46 | 4.5 |
| mean (all levels) | - | 72.6 | 5.3 |
| mean | 0 | 88.3 | 2.6 |
| mean | 1 | 68.8 | 6.05 |
| mean | 2 | 53.0 | 9.3 |

# E    DETAILS: LOW-LEVEL POLICY

## E.1    DATASET

As described in section 5.1 we train on 2000 trajectories for each of the described tasks on a simple object split using the ClevrSkills simulator. Each object color is randomly chosen from the following list: *cyan*, *red*, *white*, *yellow*, *black*, *blue*, *green*, *purple*, while the object shapes are randomly chosen from a list of *cube*, *cylinder*, *triangle*, *hexagon*. Each dataset is generated using random seeds $12000$ to $14000$ of the simulator.

## E.2    HYPERPARAMETERS

We use the AdamW optimizer for all experiments with a learning rate of $1.0e - 4$, beta values of $[0.95, 0.999]$, epsilon $1.0e - 8$, weight decay $1.0e - 6$ and a cosine learning rate scheduler. Furthermore, following the choice of Chi et al. (2023) we keep an exponential moving average (EMA) of the model weights using the same hyperparameters. However, we deviate in terms of using historical observations and proprioception values and only provide one timestep of observations into the model. Regarding the diffusion head, we use DDPM Ho et al. (2020) with standard hyperparameters:

Table 3: DDPM Noise Scheduler Hyperparameters

| Hyperparameter | Value |
|---|---|
| num_train_timesteps | 100 |
| beta_start | 0.0001 |
| beta_end | 0.02 |
| beta_schedule | `squaredcos_cap_v2` |
| variance_type | `fixed_small` |
| clip_sample | `True` |
| prediction_type | `epsilon` |

### E.3    CACHE INFO

To separate code instructions from cache information, we use a simple regular expression scanning for `pose_dict` values being set. Algorithm 1 illustrates this behavior. During inference, we can extract caching information from the VLM memory buffer by calling `ExtractMemoryInfo` on its memory of past key-step instructions $m_t$. This ensures that all instructions that attempt to assign to some key of `pose_dict` are persistent through time and visible to the low level policy in the form of $\ell^{cache}$. If no memory info is returned by the function (i.e. if none of the instructions in $m_t$ were writing to `pose_dict`), we set $\ell^{cache}$ = "null". Figure 2 illustrates the extraction of memory information from the code trace in *PlaceNextTo*. In this task, the oracle (and, as a result, the trained VLM) uses the first timestep to calculate a target placing position alongside outputting a moving instruction. This instruction is then persistent in the memory buffer $m_t$ of the VLM, which we in turn extract in the form of $\ell^{cache}$ to feed into the low level policy at every time step.

### E.4    TESTING

During inference, we use 10 denoising steps for faster inference using the DDIM sampler Song et al. (2020). We always test on 64 random initializations of the environment with seeds 10 to 74.

---

**Algorithm 1** Extract Memory Info from Python String

---
```
 1: procedure EXTRACTMEMORYINFO(python_string)
 2:     Define cache_pattern as regex: pose_dict['.*?']   =
 3:     Split python_string into lines by newline
 4:     Initialize empty list cache_lines
 5:     Initialize empty list remaining_lines
 6:     for each line in lines do
 7:         if regex_pattern matches line then
 8:             Append line to cache_lines
 9:         else
10:             Append line to remaining_lines
11:         end if
12:     end for
13:     Join remaining_lines into remaining_string
14:     Join cache_lines into cache_string
15:     return cache_string, remaining_string
16: end procedure
```
---

## F    COMPUTE RESOURCES

All of our experiments were conducted on a mixture of A100-80GB and V100-32GB GPUs. The high level VLM can be trained on a node of 8 A100s within 48 hours, while the low level policy can be trained separately and requires fewer compute resources. We trained the diffusion policy on a node of 8 V100s for around 24-72 hours depending on the size of the dataset. For our biggest dataset, we train for 250 epochs, taking around 72 hours of walltime. For inference, a single A100 is sufficient to

run both the high and lowlevel policy in parallel, i.e. they consume less than 80GB of memory in total.

## G    SOCIETAL IMPACT

Enabling learning of arbitrary robotic manipulation policies has the potential for societal impact. Our work was performed on simple environments with simple objects, thus limiting the direct potential negative impact and limiting the application to stationary robots in e.g. a warehouse setting. Nonetheless, we acknowledge that the automation of data collection and improving scalability of robot learning can have drastic societal impact due to the possibility to automate previously challenging tasks that required human supervision. This can lead to the replacement of human workers with robots. In the longer term, this can also accelerate the development of arbitrary robot policies which can be used for warfare or other malicious activities.

