# OpenReview forum: "From Code to Action: Hierarchical Learning of Diffusion-VLM Policies"
_ICLR.cc/2026/Conference — ICLR 2026 Conference Withdrawn Submission_

### Official Review · Reviewer_871p · 2025-10-31

**Soundness:** 3
**Presentation:** 2
**Contribution:** 1
**Rating:** 0
**Confidence:** 5

**Summary:**

This paper introduces a hierarchical framework for robotic imitation learning designed to improve generalization and data efficiency on complex manipulation tasks. The core idea is to decompose the learning problem into two stages: "thought imitation" and "action imitation." First, a code-generating Vision-Language Model (VLM) is trained to act as a high-level policy. This VLM learns to imitate the "thoughts" of an oracle (a scripted API-based policy) by mapping a high-level natural language instruction and visual observation to the specific, semantically meaningful API code (e.g., pick(...), place_on_actor(...)) that an expert would execute . Second, a low-level diffusion policy is trained to perform "action imitation" by learning to execute these VLM-generated code snippets to produce low-level robot actions.

**Strengths:**

The paper is well-written, and the proposed hierarchical framework is explained clearly. The core concept of "Thought Imitation" (VLM-based code generation) and "Action Imitation" (code-conditioned diffusion policy) is intuitive and well-illustrated by Figure 1.

**Weaknesses:**

1. Over-engineered and Handcrafted Framework: The entire method is critically dependent on the pre-existence of a comprehensive, scripted, open-source robotic API that can already solve the tasks. This is an extremely strong assumption that does not generalize. The framework does not learn to decompose tasks; it learns to mimic a human-engineered decomposition provided by the API (e.g., get_actor, pre_pick_ee_pose, place_on_actor, etc.) . This shifts the entire burden of task decomposition from the model to a human programmer, which is the opposite of scalable skill discovery.

2. Brittleness of the System: The approach is laden with handcrafted components that limit its applicability. For example, the system relies on a complex, rule-based memory mechanism (l^{cache}) that requires a custom regex parser (Algorithm 1) to scan the generated code strings for stateful information (e.g., pose_dict['.*?']) . This is a brittle, non-learned, and highly-engineered solution for state tracking. The system can only function within the rigid syntax of its pre-defined API, making it incapable of handling any task or object interaction not explicitly defined by that API.

3. Limited Novelty: The concept of using a VLM to generate code as a high-level plan ("Code as Policies") is well-established (e.g. as cited by the authors, Liang et al., 2023; Singh et al., 2023) . The paper's main claim to novelty is that it also learns the low-level policy via diffusion, rather than just executing the oracle API code. However, this "distillation" of a perfect, scripted oracle into a learned diffusion policy is of questionable value; the experiments (Table 1) show this learned policy (VLM+DP, 63.95% avg) is significantly worse than just executing the VLM's plan with the oracle API in the first place (VLM+Oracle, 80.78%). Furthermore, the broader concept of hierarchical policies with intermediate representations is a very active area, and the paper fails to compellingly argue why API code is a superior representation to learned latent plans (Bjorck et al., 2025) or natural language sub-goals (Shi et al., 2025).

4. Unfair Experimental Comparison: The primary baseline is a "flat" diffusion policy (DP) conditioned only on the high-level natural language prompt. This is a strawman argument. The hierarchical "VLM+DP" policy receives detailed, step-by-step, correct code instructions at each phase of the task, while the "flat" policy receives a single, ambiguous, long-horizon goal. This experiment does not prove that a hierarchical policy is better than a flat one; it proves that a policy given step-by-step guidance is better than a policy given none. A fair baseline would have been a flat policy also conditioned on VLM-generated natural language sub-goals, or a comparison to other state-of-the-art hierarchical methods.

**Questions:**

1. The framework's success seems entirely contingent on having an existing, expert-programmed, scripted API for all tasks. How do you propose this method would be applied to novel tasks or domains where no such API exists? Does this not simply shift the difficulty from data collection to manual, expert-level API engineering?

2. The paper's main novelty claim is distilling the low-level API functions into a learned diffusion policy. However, your results in Table 1 show that the performance of the full "VLM+DP" system (63.95% average) is dramatically lower than the "VLM+Oracle" system (80.78%). Given that the oracle API already exists and performs better, what is the practical motivation for replacing it with a less effective learned diffusion policy?

3. The main baseline ("Task Prompt Only (DP)") is a flat policy conditioned on a single, high-level task description. This seems like an unfair comparison against your hierarchical policy, which receives granular, step-by-step code instructions. Could you justify this baseline choice? How do you think your method would compare against a flat policy that was also given intermediate, VLM-generated natural language sub-goals?

4. The memory system for handling non-Markovian state (l^{cache}) relies on a regex parser (Algorithm 1) to find specific variable assignments in the generated code string . This seems very brittle. Why was this hand-engineered approach chosen over a more standard, learned memory mechanism, such as simply passing the VLM's hidden state to the diffusion policy?

---

### Official Review · Reviewer_KD5Y · 2025-11-01

**Soundness:** 2
**Presentation:** 1
**Contribution:** 2
**Rating:** 2
**Confidence:** 3

**Summary:**

Leverage a VLM to generate code that uses either an oracle or diffusion policy to take action in the environment. By building up a robot state in a code-cache, the model has history which is not otherwise captured in the visual state. A code-captioning objective can then be used which is presumably simpler to train than modern architectures. A simplified task space with perfect proprioception is used in the experiments.

**Strengths:**

The motivation to have hierarchy and to explore the abstractions required for robotics is a useful pursuit.

**Weaknesses:**

*Motivation*
I broadly understand the goal of induced hierarchy, particularly as it pertains to enabling long-horizon planning, and of "code as state". I however do not see the resulting hierarchy clearly explained in the actual approach which only discussed rather simple and seemingly flat tasks.

*Presentation*
I found this paper surprisingly hard to parse, having to constantly cross reference or guess at the meanings of what seem like extraneous variables in the math. Even the data used -- whose functions are essential to understanding the scope of this paper -- requires reading another paper (for environments) and the appendix (sort of, as the code listed doesn't capture what's in the figures).

*Experiments*
The design of the experiments, particularly as they differ dramatically from what is common in the field or what is used in most of the related work, needs to be better argued.  This includes L0,L1 but also the task itself. It's possible that even just having better figures for showing varying depth hierarchies and horizons or performance by trajectory length would assist here.

*Comparisons*
There are no comparisons to current models in the space (either VLAs or VLM based models) which also makes the results hard to interpret. If the idea is a focus on architecture and controlled experiments, as alluded to in the related work, then justification and verbiage aligning the results to those of other architectures should be included. For example, CLIP-RT, does not have hierarchy but is a very simple VLM style approach

**Questions:**

Please clarify the points above regarding
- why comparisons were not included
- why these tasks were appropriate
- why it was important to simplify the evaluation domain
- how the papers results will be applicable to more realistic settings.

---

### Official Review · Reviewer_HmxB · 2025-11-01

**Soundness:** 3
**Presentation:** 3
**Contribution:** 3
**Rating:** 6
**Confidence:** 3

**Summary:**

This paper presents an innovative hierarchical framework for robot policy learning, designed to address the poor generalization often seen in traditional "flat" imitation learning approaches when applied to long-horizon, compositional tasks. The core concept is realized through a two-tier architecture: 1) a high-level Vision-Language Model (VLM) that transforms natural language instructions into structured Pythonic code plans, and 2) a low-level diffusion policy that executes the generated code to produce accurate robot actions. To validate the code-as-policy concept, the authors conduct a quantitative analysis on the feature extraction capabilities. Extensive experiments on the ClevrBench benchmark demonstrate the proposed method's effectiveness in achieving compositional generalization.

**Strengths:**

1. The key insight of leveraging code as a structured intermediary representation is innovative.
2. The paper provides a good evaluation with the demonstration of compositional generalization being evidence.
3. The intuition of separating high-level planning from low-level control is impressive.

**Weaknesses:**

1. The method's effectiveness is intrinsically tied to the quality of the underlying scripted policies, raising significant questions about its transferability to real-world scenarios without such pre-existing structures.
2. The paper would be strengthened by comparisons with other recent hierarchical methods that use different intermediate representations to better establish the advantage of using code.
3. The lack of validation on a physical robot platform is one of the major limitations.

**Questions:**

Regarding the methodological and experimental design:
1. The proposed method heavily relies on a predefined API vocabulary, which may limit its generalization to novel objects or actions not covered in the API. What are the advantages of this code-based approach compared to other possible intermediate representations?
2. The memory mechanism only caches historical code instructions without verifying the actual physical state, how does the method handle potential error accumulation caused by state discrepancies?
3. Since the paper doesn't provide a thorough analysis of error propagation across the cascaded VLM-planning and policy-execution modules, how does the method clarify the bottlenecks of the entire system?

Regarding the comparisons and evaluations:
1. The generalization tests appear focused on known task compositions. How does the approach ensure rigorous validation of compositional generalization without testing zero-shot generalization to unseen task combinations?
2. Without including comparisons to state-of-the-art hierarchical methods, why is the method only compared with a flat diffusion policy baseline?
3. Given that experimental validation was conducted entirely in a simulated environment, how do the authors justify the practical applicability of the method, considering the absence of real-world robot experiments?

Overall, although the paper has major limitations in comparative benchmarking and the absence of real-world validation, it presents a novel and promising hierarchical framework. I recommend accept at this stage.

---

### Official Review · Reviewer_pnGb · 2025-11-02

**Soundness:** 2
**Presentation:** 2
**Contribution:** 2
**Rating:** 2
**Confidence:** 4

**Summary:**

This paper introduces a novel hierarchical framework in the field of robot manipulation. The first phase leverages Vision-Language Models (VLM) to generate code for robot execution. In the second phase, the code is used as a condition for the low-level diffusion policy, ultimately resulting in execution actions. The authors validate the effectiveness of their approach using the CleverSkills benchmark.

**Strengths:**

This work provides a detailed exposition of the two-phase "code to policy" paradigm. An interesting aspect is the authors' discovery that using code as a condition for diffusion can yield improved performance. To address non-Markovian tasks, the authors designed a memory module to record historical robot states. Additionally, they introduced an auxiliary loss to assist in the training of Vision-Language Models (VLMs). The overall pipeline of the method is well-structured and clear.

**Weaknesses:**

1.The core issue lies in the motivation. The authors argue that an oracle policy inherently possesses the ability to decompose tasks, allowing for the segmentation into multiple subtasks without additional annotation. However, this ability seems to be limited to the settings provided by the ClevrSkills benchmark proposed by the authors. What about benchmarks that lack such an oracle policy, like the commonly used manipulation simulation benchmarks Calvin[1] and Libero[2]?
2.The paper lacks experiments with real-world robots. In real environments, even using code, manual annotation remains necessary, which seems to contradict the authors' claim that an oracle policy reduces annotation costs. This step appears unavoidable. Additionally, even in ClevrSkills, the authors mention the need to filter out unsuccessful trajectories, which seemingly increases the manual annotation workload.
3.The experiments are insufficient. It would be valuable to compare this code-layering paradigm, which integrates sub-task language instructions into diffusion policy (dp), with alternatives. This comparison could demonstrate that using code as a condition is more easily learned by the dp for low-level policy compared to using sub-task language instructions as a condition.

[1]CALVIN: A Benchmark for Language-Conditioned  Policy Learning for Long-Horizon Robot  Manipulation Tasks
[2]LIBERO: Benchmarking Knowledge Transfer for Lifelong Robot Learning

**Questions:**

Please see the weakness.

---

### Note · Authors · 2025-11-12

**Comment:**

We would like to withdraw our submission. While we appreciate the time invested by the reviewers, we believe the reviews did not accurately reflect the motivation or contribution of our work. Several critiques focused on assumptions that were either explicitly addressed in the paper or are common in related literature, while overlooking the core research question we aimed to explore.

Some comparisons requested by reviewers (e.g., against unrelated architectures or baselines that do not share our design goals) would not have been fully meaningful within the scope of our study. Our intention was to isolate and analyze the architectural implications of hierarchy and code as state, not to compete on raw performance in uncontrolled settings. Unfortunately, this distinction was not recognized in the feedback.

We will take this opportunity to revise the manuscript to better clarify the motivation, strengthen the experimental design, and explicitly position our work relative to existing approaches.

Thank you for your consideration.

**Withdrawal Confirmation:**

I have read and agree with the venue's withdrawal policy on behalf of myself and my co-authors.